**American Society for Microbiology** | **Microbiology Spectrum**

ⓐ | **Open Peer Review** | Bacteriophages | Research Article

# Novel lytic phages improve the antibiofilm activity of dalbavancin, daptomycin, and fosfomycin against vancomycin-resistant enterococci

Rima Fanaei Pirlar,[1] Alvaro Irigoyen-von-Sierakowski,[1,2] Jeroen Wagemans,[3] Yu Ning,[1] Rob Lavigne,[3] Andrej Trampuz,[4,5] Svetlana Karbysheva[1]

**ABSTRACT** Periprosthetic joint infections (PJI) caused by *Enterococcus* spp., especially vancomycin-resistant strains (VRE), are challenging to treat due to biofilm tolerance and limited antibiotic options. Bacteriophages offer a promising adjunct through targeted and biofilm-disrupting activity. This study evaluated two novel lytic phages, alone and combined with last-line antibiotics, for their ability to eradicate VRE biofilms *in vitro*. Two novel lytic phages, CUB-FM (*E. faecium*) and CUB-FS (*E. faecalis*), were isolated from hospital sewage and characterized via whole-genome sequencing and transmission electron microscopy. Antibiofilm efficacy of phages alone and in combination with dalbavancin, daptomycin, and fosfomycin was assessed against biofilm-embedded VRE strains using isothermal microcalorimetry. Synergy was defined as a combined effect exceeding the sum of individual activities. Genomic analysis confirmed both phages as strictly lytic and free of lysogeny, virulence, or resistance genes. TEM classified CUB-FM within *Salasmaviridae* and CUB-FS within *Herelleviridae*. Both exhibited dose-dependent antibiofilm activity, with optimal efficacy at $10^{12}$ (CUB-FM) and $10^8$ PFU/mL (CUB-FS). While antibiotic monotherapies showed limited antibiofilm effects, phage-antibiotic combinations markedly enhanced activity. CUB-FM with dalbavancin achieved the strongest suppression against *E. faecium* ($t_{Max}$22.3 h vs. 5.2 h control, $P < 0.001$), and CUB-FS with dalbavancin or fosfomycin at ≥10 × MIC completely eradicated *E. faecalis* biofilms. Daptomycin-phage combinations produced additive to synergistic effects. Novel phages CUB-FM and CUB-FS exhibit potent antibiofilm activity and synergize with last-line antibiotics against VRE. Phage-antibiotic combinations, particularly with dalbavancin and fosfomycin, represent a promising strategy for treating biofilm-associated enterococcal PJIs.

**IMPORTANCE** Vancomycin-resistant enterococci (VRE) are increasingly implicated in biofilm-associated periprosthetic joint infections, where treatment options are limited, and clinical outcomes are poor. Conventional antibiotics often fail due to reduced biofilm penetration and bacterial tolerance, highlighting the need for novel therapeutic strategies. Our study introduces two newly characterized lytic phages, CUB-FM and CUB-FS, which demonstrated strong antibiofilm activity and synergistic interactions with last-line antibiotics. Notably, phage-antibiotic combinations achieved either additive or synergistic effects, with dalbavancin and fosfomycin-phage therapy leading to a complete eradication of *E. faecalis* biofilms. These findings provide proof of concept that combining phages with antibiotics enhances efficacy against multidrug-resistant *Enterococcus* biofilms, offering a translational pathway for personalized, adjunctive therapies in complex orthopedic infections. By bridging the gap between genomic phage safety validation and functional synergy testing, this work supports

Address correspondence to Svetlana Karbysheva, svetlana.karbysheva@charite.de.

Rima Fanaei Pirlar and Alvaro Irigoyen-von-Sierakowski contributed equally to this article. Author order was determined by mutual agreement after discussion among the contributing authors.

The authors declare no conflict of interest.

further preclinical and clinical development of phage-antibiotic strategies for refractory implant-associated infections.

**KEYWORDS** bacteriophages, enterococci, VRE, antibiotics, periprosthetic joint infection, biofilms

Periprosthetic joint infections (PJI) represent a severe complication in orthopedic surgery, significantly affecting patient outcomes and healthcare systems. Among the diverse array of pathogens implicated, *Enterococcus* spp. account for 2 to 15% of cases and pose notable challenges due to their ability to form biofilms on prosthetic surfaces (1, 2). Biofilms not only shield bacteria from the host immune response but also significantly reduce the efficacy of conventional antibiotic therapies (3).

The clinical management of enterococcal-related PJI is further complicated by the emergence of vancomycin-resistant *Enterococcus* spp. (VRE), which restricts treatment options to reserve antibiotics, such as glycopeptides, oxazolidinones, or lipopeptides (4, 5). This growing resistance highlights an urgent need for alternative therapeutic approaches.

Bacteriophages (phages), viruses that specifically target bacteria, have re-emerged as a promising tool against biofilm-associated infections (6–8). Their specificity, self-amplifying nature, and biofilm-degrading capacity make them an innovative therapeutic tool for treating persistent infections associated with medical devices. Recent studies have demonstrated that lytic phages can penetrate and disrupt biofilms on orthopedic implants, reducing bacterial load and delaying regrowth (8, 9). In the context of PJI, phage therapy may offer a minimally invasive adjunct or salvage option when surgical debridement and systemic antibiotics fail (8). Importantly, phages can also act synergistically with antibiotics, offering the potential to further improve treatment efficacy (10).

Although phage-antibiotic combinations have shown promise against VRE biofilms, for instance, complete eradication of *E. faecalis* biofilms with phage φEFP01 plus vancomycin (11), comprehensive data remain limited. However, most studies have focused on planktonic cultures or a limited set of antibiotics (10), leaving a gap in knowledge regarding phage synergy with clinically relevant agents, such as dalbavancin, daptomycin, and fosfomycin, against biofilm-forming VRE.

In this context, our study aims to address this gap by isolating novel VRE-targeting phages and evaluating their antibiofilm activity in combination with last-line antibiotics, exploring viable therapeutic strategies for challenging multidrug-resistant enterococcal infections in orthopedic settings.

## RESULTS

### MIC determination of antibiotics and genomic analysis of *Enterococcus* strains

The phenotypic and genotypic profiles of both strains demonstrated that all were MDR, showing non-susceptibility to at least one agent in three or more antimicrobial categories. The MIC profiles of both isolates reflected their genotypic resistance backgrounds. Vancomycin resistance (≥32 µg/mL) corresponded with the presence of the *vanHBX* operon in both strains, while their susceptibility to teicoplanin (MIC < 0.05 µg/mL) aligns with the known phenotype of vanB-type resistance, which typically confers resistance to vancomycin but not teicoplanin. Additionally, *E. faecalis* displayed high-level gentamicin resistance, consistent with the presence of aminoglycoside-modifying enzymes (*ant (6)-Ia* and *aph(3')-III*), whereas *E. faecium*, which carried *aac(6')-Ii*, remained below the high-level resistance breakpoint. Levofloxacin resistance in *E. faecium* could not be attributed to acquired resistance genes, indicating the likely involvement of chromosomal mutations. In contrast, all other tested agents (linezolid, tigecycline, fosfomycin, dalbavancin, and daptomycin) displayed phenotypic susceptibility consistent with genomic predictions (Table 1; Fig. 1). Whole-genome sequencing (WGS) revealed distinct virulence gene repertoires in *E. faecium* and *E. faecalis* (Fig. 2).

**TABLE 1** MIC values of different antibiotics against *E. faecium* and *E. faecalis* (in µg/mL)[a]

| Strain | Vancomycin, µg/mL (CLSI/EUCAST) | Teicoplanin, µg/mL (CLSI/EUCAST) | HLG, 500 µg/mL (CLSI/EUCAST) | Tigecycline µg/mL (CLSI/EUCAST) | Linezolid, µg/mL (CLSI/EUCAST) | Levofloxacin, µg/mL (CLSI/EUCAST) | Fosfomycin, µg/mL (CLSI/EUCAST) | Daptomycin, µg/mL (CLSI/EUCAST) | Dalbavancin, µg/mL (CLSI/EUCAST) |
|---|---|---|---|---|---|---|---|---|---|
| *E. faecium* | ≥32 (R/R) | <0.05 (S/S) | <500 (S/S) | <0.12 (S/S) | 2.0 (S/S) | ≥8 (R/R) | 64.0 (S/S) | 0.5 (S/S) | 0.023 (S/S) |
| *E. faecalis* | ≥32 (R/R) | <0.05 (S/S) | >500 (R/R) | <0.12 (S/S) | 1.0 (S/S) | 0.5 (S/S) | 48.0 (S/S) | 0.5 (S/S) | 0.125 (S/S) |

[a]S, susceptible; VRE, vancomycin-resistant *Enterococcus* spp.; HLG, high-level gentamicin; CLSI, Clinical and Laboratory Standards Institute; EUCAST, European Committee on Antimicrobial Susceptibility Testing. *E. faecium* and *E. faecalis* were categorized as susceptible or resistant according to either CLSI or EUCAST breakpoints or ECOFF (epidemiological cut-off value).

The *E. faecium* isolate carried a relatively limited set of virulence determinants, including adhesins (*acm*, *ecbA*, *scm*), the aggregation factor *fss3*, and the immune evasion gene *sgrA*. In contrast, *E. faecalis* harbored a broader spectrum of virulence-associated genes across multiple functional categories. These included adhesins (*ace*, *efaA*, *EF0485*), pili/aggregation genes (*ebpA- C*, *srtC*, *EF0149*, *fss2-3*, *prgB/asc10*, *fss1*), and biofilm-related loci (*sprE*, *fsrA-C*, *bopD*, *EF0818*). Additional determinants comprised immune evasion genes (*cpsA - K*), proteases (*gelE*, *EF3023*), and the cytolysin operon (*cylR1-R2*, *cylA*, *cylS-M*). PHASTER analysis further identified one intact prophage region in *E. faecium* and two intact prophage regions in *E. faecalis*. Together, these findings highlight the higher virulence potential of the *E. faecalis* isolate, particularly in traits linked to biofilm formation, toxin production, and immune evasion.

## Genomic characterization of isolated phages

The isolated phages, CUB-FM targeting *E. faecium* and CUB-FS targeting *E. faecalis*, were characterized by WGS to determine their suitability for phage therapy. CUB-FM has a short dsDNA genome of 18,153 bp and is closely related to *Enterococcus* phage Athos (National Center for Biotechnology Information [NCBI] accession number LR990834; BLASTn analysis: 99% coverage and 92.65% sequence identity). The intergenomic similarities were calculated, revealing that CUB-FM belongs to the same unclassified genus within the *Salasmaviridae* family (Fig. S1). After genome annotation, neither indications for a lysogenic lifecycle nor antibiotic resistance and virulence genes were found. The strictly lytic nature of this phage was further validated using Phage.AI. Phage CUB-FS contains a 142,310 bp dsDNA genome and is related to *Enterococcus* phage phiEF24C (NCBI accession number NC_009904; BLASTn analysis: 91% coverage and 98.56% sequence identity). The intergenomic distance between these related viruses showed CUB-FS represents a new species within the *Kochikohdavirus* of the *Brockvirinae* within the *Herelleviridae* family (Fig. S1). It also does not contain any unwanted genes and can be concluded as a lytic phage.

## Transmission electron microscopy (TEM) visualization of isolated phages

TEM analysis of phage CUB-FM revealed a slightly elongated icosahedral head approximately 47 nm in diameter and a short tail measuring about 19 nm in length (Fig. 3A through C). The tail appeared non-contractile and lacked visible tail fibers, consistent with members of the *Salasmaviridae* family. The CUB-FS displays an icosahedral head (77 nm in diameter) and a contractile tail (197 nm in length, 20 nm in width) with no visible tail fibers (Fig. 3D through F). A baseplate structure was observed at the distal end of the tail, supporting classification within this family. These features, together with genomic analysis, confirm that CUB-FS belongs to the *Herelleviridae* family.

## Phage efficacy testing

Phages propagated on *E. faecium* as well as *E. faecalis* consistently reached exceptionally high concentrations of around $4 \times 10^{13}$ PFU/mL. For *E. faecium*, $10^{12}$ PFU/mL was identified as the most effective concentration (Fig. 4A). The mean bacterial heat

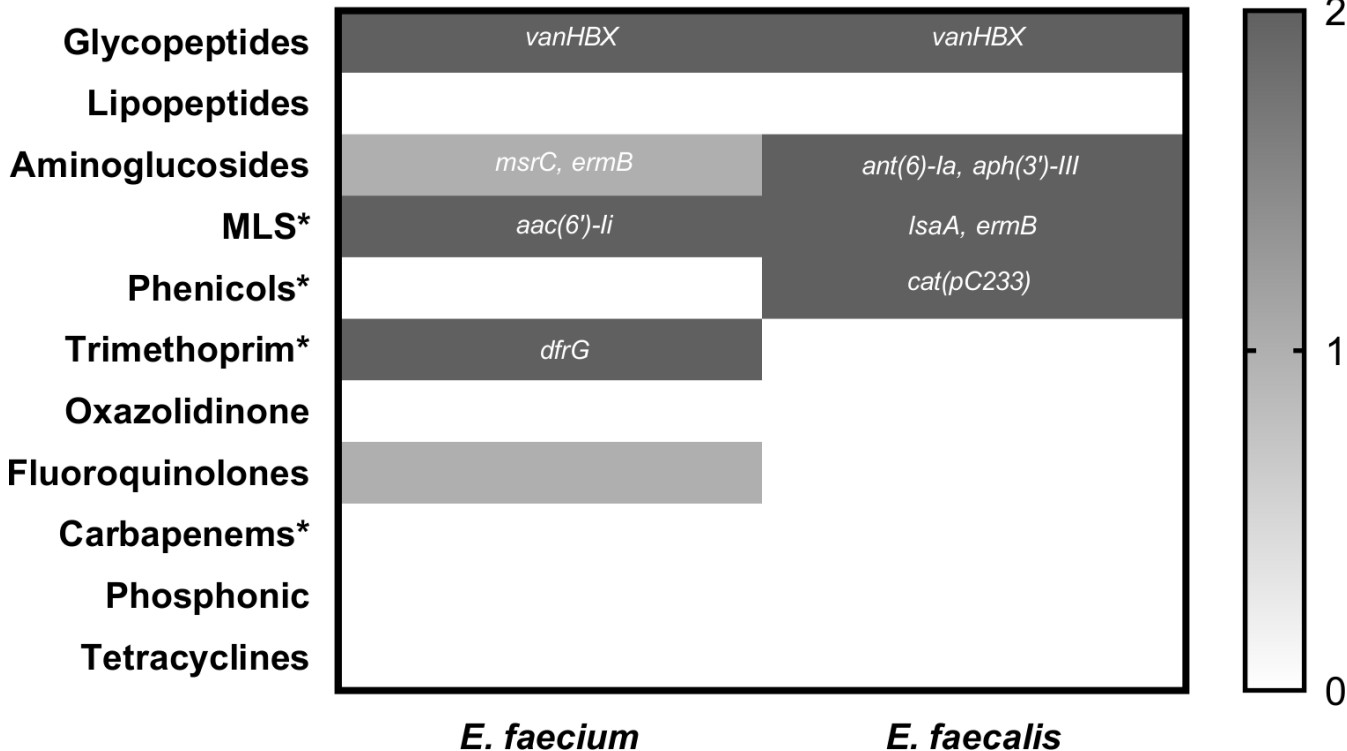

**FIG 1** Correlation of genomic predictions with phenotypic resistance profiles in VRE strains. Heatmap illustrating the relationship between WGS-predicted resistance determinants and MIC-based susceptibility results for *E. faecium* and *E. faecalis*. The scale denotes the level of correlation: 2 = gene present and phenotype confirmed; 1 = inconsistent results between genotype and phenotype; and 0 = gene absent and phenotype confirmed. Both strains carried the vanHBX operon, correlating with vancomycin resistance and teicoplanin and dalbavancin susceptibility consistent with a vanB-type phenotype. High-level gentamicin resistance was observed in *E. faecalis*, matching the presence of *ant (6)-Ia* and *aph(3')-III*, whereas *E. faecium* carried *aac(6')-Ii* but remained phenotypically susceptible. Levofloxacin resistance in *E. faecium* was not explained by acquired resistance genes, suggesting chromosomal mutations. All other agents (linezolid, tigecycline, fosfomycin, dalbavancin, and daptomycin) showed phenotypic susceptibility in line with genomic predictions. *, MIC was not determined; MLS, macrolides/lincosamides/streptogramins.

detection time following treatment with $10^{12}$ PFU/mL of phages was significantly prolonged (15.6 h) compared to GC (7.4 h), $P = 0.008$. Further increasing the phage titer to $10^{13}$ PFU/mL did not provide additional benefit, as the heat detection time decreased to 9.1 h. In contrast, for *E. faecalis*, at lower phage concentrations ($10^5$–$10^7$ PFU/mL), heat flow curves displayed an early metabolic peak (~1.8 h), followed by regrowth (~11.6 h), reflecting partial suppression and rebound of the biofilm population. In contrast, higher titers ($10^8$ and $10^9$ PFU/mL) suppressed bacterial metabolism from the start (near-baseline heat flow) and significantly delayed peak activity (7.0 and 6.3 h, respectively) compared to GC (3.3 h), $P < 0.01$ for both, indicating the most effective antibiofilm activity at these concentrations. There was no significant difference when the phage titer was further increased to $10^{10}$ PFU/mL (4.7 h), $P = 0.08$ (Fig. 4B). Therefore, based on previous studies and commonly used phage titers in therapeutic applications (6, 12), a dose of $10^8$ PFU/mL was selected for further analysis.

## Phage-antibiotic synergistic effect against biofilm bacteria

Antibiotics alone exhibited dose-dependent antibiofilm activity. For *E. faecium*, dalbavancin at 100× MIC achieved the strongest effect compared to the growth control, reducing $HF_{Max}$ from 210.4 ± 6.5 µW (growth control) to 86.8 ± 1.7 µW and increasing $t_{Max}$ from 5.2 ± 1.0 h to 19.0 ± 1.8 h ($P < 0.001$). Daptomycin and fosfomycin at 100× MIC also led to significant regrowth delays ($t_{Max}$: 12.5 ± 1.2 and 11.9 ± 1.7 h, respectively; both $P < 0.001$). In contrast, lower concentrations (1× MIC) of fosfomycin or daptomycin showed

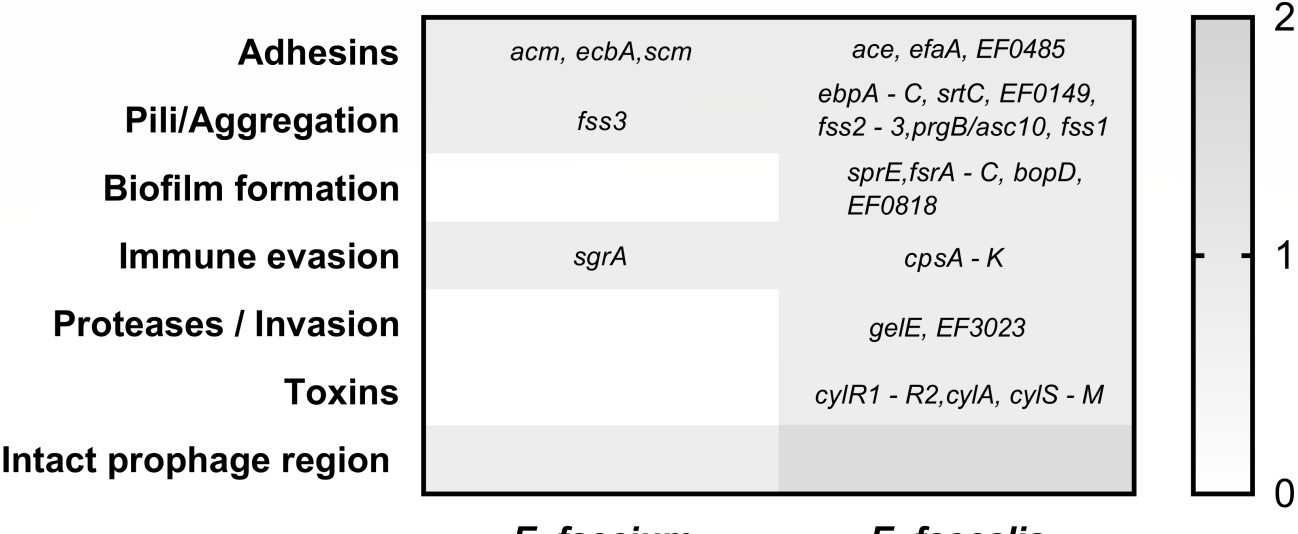

**FIG 2** Distribution of virulence-associated genes in *E. faecium* and *E. faecalis*. Heatmap representation of virulence determinants identified by WGS in the two VRE isolates. Genes are grouped by functional categories: adhesins, pili/aggregation, biofilm formation, immune evasion, proteases/invasion, and toxins. The scale bar indicates: 1 = gene presence, 0 = gene absence. *E. faecium* harbored adhesin genes (*acm*, *ecbA*, *scm*), the aggregation factor (*fss3*), and immune evasion gene (*sgrA*), whereas *E. faecalis* carried a broader set of virulence determinants, including multiple adhesins (*ace*, *efaA*, *EF0485*), pili genes (*ebpA-C*, *srtC*, *EF0149*, *fss2-3*, *prgB/asc10*, *fss1*), biofilm-associated factors (*sprE*, *fsrA-C*, *bopD*, *EF0818*), immune evasion (*cpsA-K*), proteases (*gelE*, *EF3023*), and cytolysin toxin genes (*cylR1-R2*, *cylA*, *cylS-M*). PHASTER analysis identified one intact prophage region in *E. faecium* and two intact prophage regions in *E. faecalis*.

limited efficacy, with $t_{Max}$ values not significantly different from the control. In *E. faecalis*, only high concentrations of antibiotics demonstrated notable activity. Dalbavancin at 100× MIC delayed $t_{Max}$ from 3.6 ± 1.1 to 15.0 ± 1.5 h ($P < 0.001$), while fosfomycin and daptomycin at the same concentration yielded $t_{Max}$ values of 6.0 ± 0.0 h ($P = 0.007$) and 10.6 ± 1.2 h ($P = 0.002$), respectively (Fig. 5; Table S1; Fig. S2).

Combining phages with antibiotics significantly enhanced antibiofilm activity across both species. In *E. faecium*, the combination of CUB-FM with all antibiotics produced an additive effect, with activity greater than either treatment alone but not exceeding the expected sum of their individual effects. The combination of dalbavancin at 100× MIC with CUB-FM yielded the most potent suppression ($HF_{Max}$: 45.0 ± 7.6 µW; $t_{Max}$: 22.3 ± 1.3 h; $P < 0.001$), exceeding either agent alone. Similar additive effects were observed with daptomycin and fosfomycin: $t_{Max}$ increased to 20.6 ± 1.4 and 14.6 ± 1.5 h, respectively, when combined with CUB-FM ($P < 0.001$ for both). In E. *faecalis*, complete biofilm eradication (no heat flow detected) was achieved when CUB-FS was combined with dalbavancin or fosfomycin at 10× and 100× MIC. Notably, even at 1× MIC, these combinations showed synergistic effect ($\Delta t_{Max}$ 4.1 h and 10.2, respectively), highlighting the potentiating effect of phage treatment. Daptomycin at 1× MIC combined with phage produced an additive effect, with activity greater than either treatment alone but not exceeding the expected sum of their individual effects, whereas at 10× and 100× MIC showed synergy ($\Delta t_{Max}$ 7.4 h and 22.1, respectively), significantly reducing biofilm metabolic activity beyond either monotherapy. $HF_{Max}$ values in combination therapy also showed consistent and significant reductions compared to monotherapies (Fig. 5; Table S1; Fig. S2).

## DISCUSSION

This study addresses a critical need for novel strategies against vancomycin-resistant *Enterococcus* spp., particularly in biofilm-associated infections, such as PJIs. We isolated two novel bacteriophages targeting *E. faecium* and *E. faecalis* and evaluated their therapeutic potential both alone and in combination with antibiotics commonly used for recalcitrant gram-positive infections: dalbavancin, daptomycin, and fosfomycin.

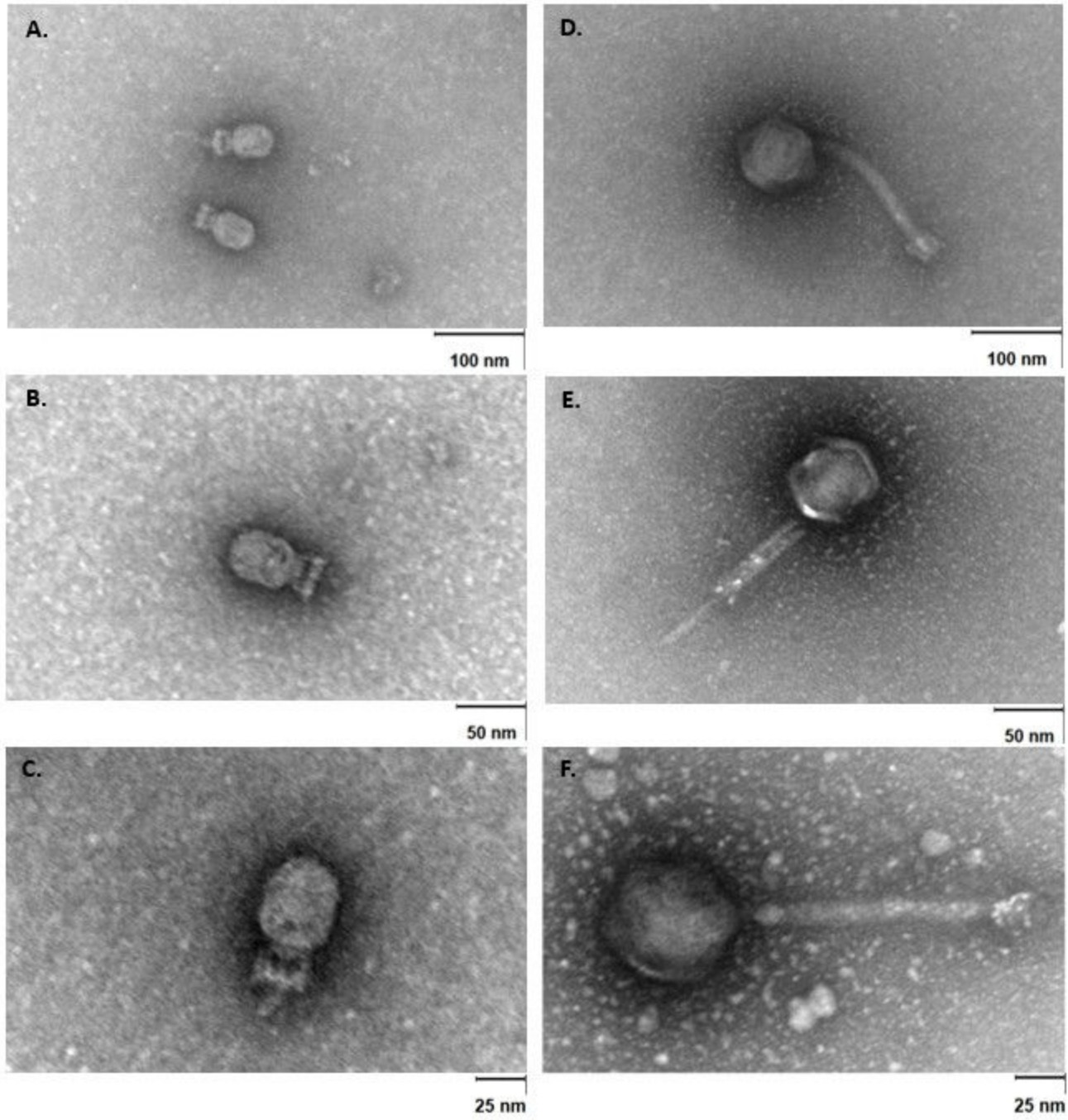

**FIG 3** TEM of the lytic enterococcal phages characterized in this study. (A–C) show phage CUB-FM infecting *E. faecium* at different magnifications with scale bars of 100 (A), 50 (B), and 25 nm (C). (D–F) show phage CUB-FS infecting *E. faecalis* at 100 (D), 50 (E), and 25 nm (F). Phage suspensions were applied to carbon-coated, glow-discharged Ni-mesh grids and negatively stained with 1% (w/v) uranyl acetate. Images were acquired using a Zeiss EM 906 operating at 80 kV. Intact virions with icosahedral heads and visible tails are clearly discernible.

In selecting antibiotics for this study, we aimed to cover diverse mechanisms of action while including agents of clinical relevance in the management of multidrug-resistant gram-positive infections. Dalbavancin, daptomycin, and fosfomycin were chosen because they represent last-line therapeutic options in VRE-associated infections, have distinct targets (cell wall synthesis, membrane depolarization, and peptidoglycan precursor inhibition, respectively) (13), and are used in clinical scenarios where phage therapy might be considered as an adjunct (14, 15). Although dalbavancin as a

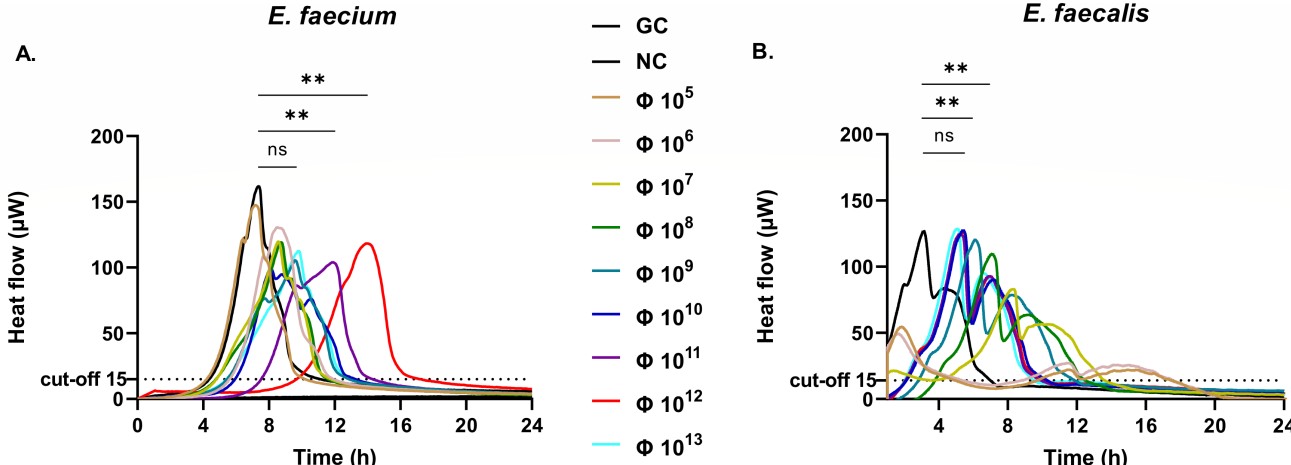

**FIG 4** Microcalorimetry analysis of *E. faecium* (A) and *E. faecalis* (B) treated with corresponding phages at different concentrations, from $10^5$ to $10^{13}$ PFU/mL. Dashed lines represent time to 15 µW heat flow (i.e., thermal growth onset time). For *E. faecalis* at lower phage concentrations ($10^5$–$10^7$ PFU/mL), heat flow curves displayed an early metabolic peak (~1.8 h), followed by regrowth (~11.6 h), reflecting partial suppression and rebound of the biofilm population. In contrast, higher titers ($10^8$ PFU/mL) suppressed bacterial metabolism from the start (near-baseline heat flow) and markedly delayed peak activity, indicating the most effective antibiofilm activity at these concentrations. Φ, phage (CUB-FM for *E. faecium* and CUB-FS for *E. faecalis*). **, $P < 0.01$.

lipoglycopeptide is mechanistically related to vancomycin, its inclusion allowed us to evaluate whether differences in binding affinity and spectrum could yield divergent activity against vanB-type VRE. As confirmed by MIC testing, both isolates demonstrated high-level vancomycin resistance consistent with their genotype, underscoring the need to explore alternative or combination strategies. In this study, antibiotic concentrations up to 100× MIC were applied in order to probe the upper range of antibiofilm activity. While such concentrations clearly exceed those achievable through systemic administration, they serve to illustrate the marked tolerance of biofilm-embedded bacteria compared to their planktonic counterparts. Importantly, these high concentrations are not without clinical relevance since comparable levels can be achieved through local delivery strategies, such as antibiotic-loaded spacers, beads, or bone cement, which provide high local drug release without systemic toxicity (16, 17). Thus, although 100× MIC data should be interpreted cautiously, they provide valuable insight into the potential limits of antibiotic efficacy within biofilms and emphasize the need for combination approaches, including phage therapy, to overcome biofilm-associated tolerance.

Antibiotic monotherapies exhibited variable antibiofilm efficacy with notable discrepancies compared to MIC-based susceptibility profiles. While MIC testing indicated susceptibility to fosfomycin, daptomycin, and dalbavancin under planktonic conditions, their activity was markedly reduced in the biofilm model. This highlights the known limitation of MIC assays in predicting efficacy against biofilm-associated infections, where bacterial communities exhibit increased tolerance due to physical barriers and altered metabolic states (18, 19). Among the agents tested, dalbavancin demonstrated comparable activity in both MIC and biofilm assays, where at 1× MIC, it significantly delayed bacterial growth of *E. faecium* compared to growth control. This aligns with previous studies reporting dalbavancin's capacity to inhibit biofilm formation and reduce biomass in gram-positive pathogens (13, 20). Daptomycin and fosfomycin also exhibited dose-dependent antibiofilm activity against *E. faecium*, with significant growth delays observed starting from 10× MIC. However, both showed minimal antibiofilm activity against *E. faecalis* biofilms. While daptomycin is established in the treatment of bloodstream infections and endocarditis, its antibiofilm efficacy may be limited by its calcium-dependent mechanism of membrane depolarization (21). Fosfomycin monotherapy has also limited efficacy against mature biofilms potentially due to suboptimal biofilm matrix accumulation and resistance development when used alone (22). Nonetheless, its

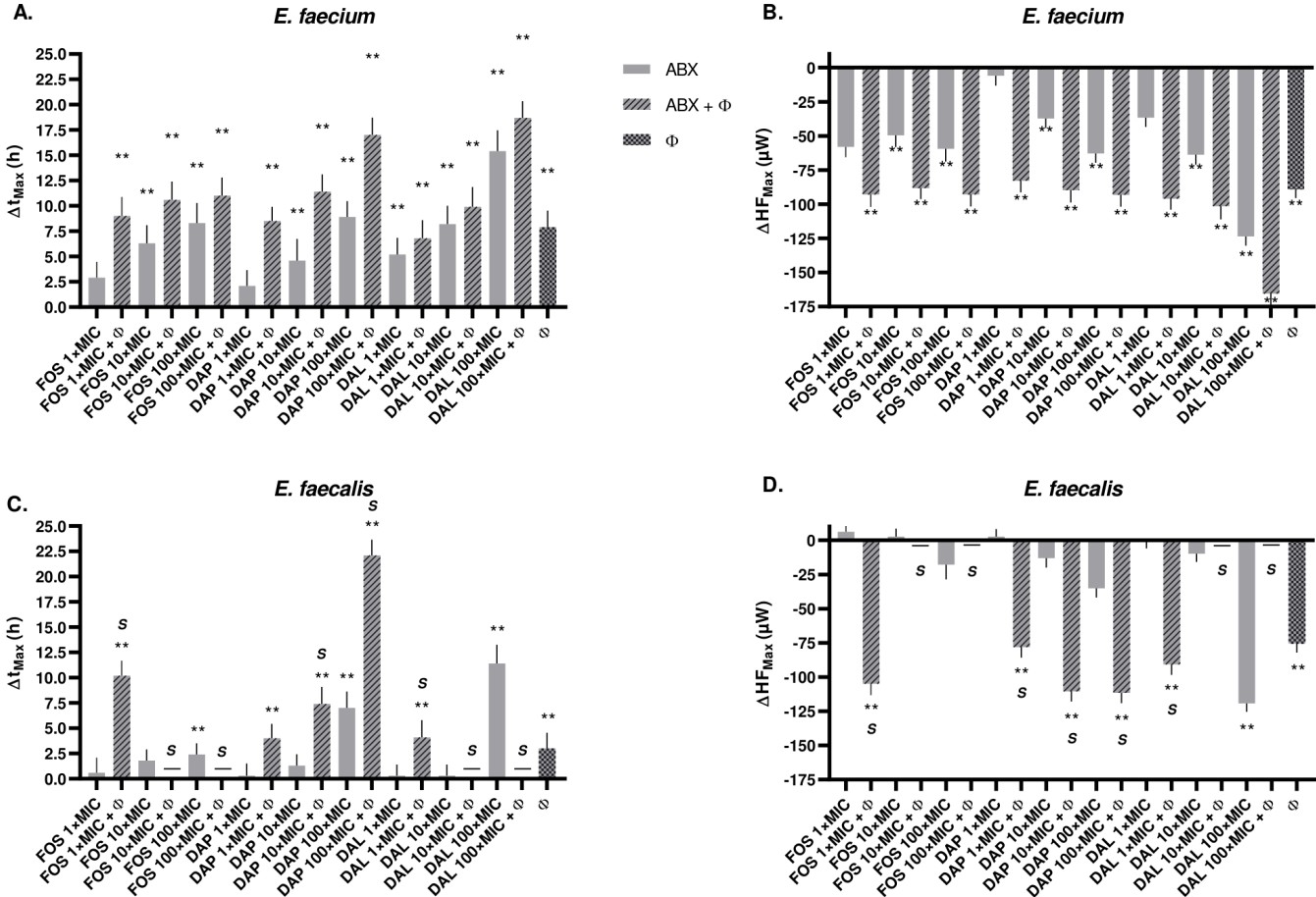

**FIG 5** Effect of combined antibiotic-phage treatments on *E. faecium* and *E. faecalis* biofilms. FOS, fosfomycin; DAP, daptomycin; DAL, dalbavancin; Φ, phage (CUB-FM for *E. faecium* and CUB-FS for *E. faecalis*); ABX, antibiotics; *S*, synergy; −, complete biofilm eradication (no heat flow detected); **, $P < 0.01$. Bar graphs show mean ± SD changes in (A) $t_{Max}$ and (B) $HF_{Max}$ in *E. faecium* and (C) $t_{Max}$ and (D) $HF_{Max}$ in *E. faecalis* relative to untreated growth controls following combined antibiotic-phage treatment at indicated multiples of MIC. Conditions include fosfomycin + phage, daptomycin + phage, and dalbavancin + phage or corresponding phage monotherapies. Positive $\Delta t_{Max}$ values indicate delayed time to maximum heat flow; negative $\Delta HF_{Max}$ values indicate reduced metabolic activity. Data are from biological triplicates. One-way ANOVA with Tukey's post-hoc test was used. Synergy is defined as a combined effect exceeding the sum of the effects of the individual agents, whereas additive effects represent improved activity over monotherapy but within the expected sum of individual effects.

favorable pharmacokinetics, including low molecular weight and effective tissue diffusion, suggest its value as part of combination regimens, rather than a standalone agent (23).

Although certain antibiotics demonstrated measurable antibiofilm activity in our assays, their efficacy was incomplete and, in several cases, markedly reduced compared to planktonic conditions. The rationale for introducing phages lies in their complementary advantages: phages can actively degrade biofilm structure, replicate at the site of infection, and limit the emergence of resistance by applying a second, independent mechanism of bacterial killing. Thus, rather than replacing active antibiotics, phages provide an additional layer of activity that enhances and prolongs antibiotic efficacy, particularly in the context of biofilm-associated VRE infections where treatment options remain limited.

In parallel, the genomic characterization of the two isolates revealed distinct virulence gene repertoires, with *E. faecium* carrying a limited set of adhesins and immune evasion genes, while *E. faecalis* harbored a broader array of adhesins, pili, biofilm-associated loci, and cytolysin genes. This more extensive virulence profile in *E. faecalis* is consistent with its stronger biofilm tolerance and higher pathogenic potential. Importantly, these findings aligned with the MIC data: both isolates displayed high-level vancomycin

resistance in agreement with their van operons, while *E. faecalis* additionally showed high-level gentamicin resistance corresponding to aminoglycoside-modifying enzyme genes detected by WGS. Such genotype-phenotype concordance underscores the value of WGS as a predictive tool while reinforcing the importance of phenotypic validation in antimicrobial testing.

For phage characterization, we prioritized genomic analysis to assess therapeutic safety. Genomic screening provides important insights into phage biology, including genome size, open reading frames, and the absence of genes associated with virulence or antimicrobial resistance, which are crucial for clinical safety. Moreover, conserved and divergent genomic regions identified by phylogenetic analyses help classify phages and guide their therapeutic selection. Equally important is the exclusion of harmful genes while recognizing beneficial features, such as anti-CRISPR loci, which may enhance phage efficacy by overcoming bacterial defense systems. Together, these genomic insights support the rationale for developing safe and effective phages against multidrug-resistant *Enterococcus* spp. (10).

Our analyses confirmed both isolated phages as suitable candidates. CUB-FM (92.65% identity to *Enterococcus* phage Athos, *Salasmaviridae*) showed a strictly lytic profile without lysogeny, virulence, or resistance genes, and TEM confirmed a podovirus morphology with an icosahedral head and a short non-contractile tail (24, 25). CUB-FS showed high sequence identity (98.56%) to phage phiEF24C, within the *Herelleviridae* family. Like CUB-FM, CUB-FS lacked lysogeny- or resistance-associated genes and displayed classic *Herelleviridae* morphology with an icosahedral head and a contractile tail. TEM confirmed the presence of a baseplate structure.

Interestingly, phages propagated on *E. faecium* and *E. faecalis* consistently reached higher titers (~$4 \times 10^{13}$ PFU/mL). However, in the efficacy assays, different titers were identified for different bacterial strains, and relatively high phage titers were employed to ensure reproducible activityPEER REVIEW HISTORY against biofilm-embedded *E. faecium*. This discrepancy likely reflects differences in host-phage interactions, including replication dynamics, burst size, and adsorption efficiency (26, 27). Such host-dependent variability in phage yield has been described previously and highlights the importance of host selection in achieving high-titer preparations for therapeutic applications (28). In the efficacy assays, relatively high phage titers were employed to ensure reproducible activity against biofilm-embedded bacteria. This approach reflects the well-recognized challenge that biofilms pose as a physical and physiological barrier, often necessitating higher phage inputs compared to planktonic cultures. While such concentrations may exceed those typically achievable with systemic administration, they are relevant for therapeutic strategies based on local delivery. Intra-articular instillation, intraoperative irrigation, or the use of phage-loaded biomaterials can yield locally elevated titers without systemic toxicity, as already reported in preclinical and compassionate-use settings (8). Thus, the high titers used *in vitro* should be viewed as proof-of-concept conditions to demonstrate antibiofilm efficacy. Future studies are warranted to establish the minimal effective dose under clinically achievable conditions and to optimize phage-antibiotic combinations for translation into therapeutic applications.

In microcalorimetry assays, CUB-FM exhibited optimal biofilm suppression at $10^{12}$ PFU/mL. Interestingly, higher titers ($10^{13}$ PFU/mL) paradoxically decreased efficacy, potentially due to lysis inhibition or quorum sensing interference at high phage densities (29–31). CUB-FM, thus, emerges as a promising therapeutic candidate, demonstrating genomic safety and optimal performance at mid-to-high phage concentrations.

Similarly, antibiofilm assays with CUB-FS revealed a plateau effect beyond $10^8$ PFU/mL, suggesting phage saturation or limited penetration into dense biofilm matrices. This aligns with previous observations indicating diminishing returns at higher phage titers in mature biofilms (9, 32). CUB-FS, therefore, shows strong therapeutic potential at clinically relevant titers, supported by a favorable safety profile and consistent morphological features. However, due to the observed saturation effect, its efficacy may be

better enhanced through combination with antibiotics or biofilm-disrupting agents rather than by increasing phage dose alone.

Furthermore, we evaluated the synergy between antibiotics and phages, which can lower required drug doses, reducing toxicity and delaying resistance by forcing bacteria to overcome multiple mechanisms simultaneously (33, 34). This is particularly relevant in biofilm-associated and multidrug-resistant infections, where phages enhance antibiotic penetration and disrupt biofilm tolerance (35). In our study, all phage-antibiotic combinations produced additive or synergistic effects, supported by calorimetric data showing significant reductions in $HF_{Max}$ and extensions of $t_{Max}$ compared to monotherapies. The combination of dalbavancin and phages yielded the most potent biofilm suppression. The strongest effect was observed for dalbavancin with phage CUB-FS in *E. faecalis*, where synergy at 1× MIC and complete biofilm clearance at higher concentrations were achieved, consistent with recent clinical reports (14). For *E. faecium*, the same combination showed additive effects, still outperforming dalbavancin alone. Although direct *in vitro* evidence on dalbavancin-phage combinations is limited, our results indicate that this strategy effectively inhibited biofilm formation. The underlying mechanisms likely include a reduction in the effective antibiotic concentration, enhanced drug penetration into the biofilm matrix via phage-encoded depolymerases, and suppression of resistance development, mechanisms that are probably similar to those described for vancomycin (36, 37). Daptomycin-phage combinations yielded additive activity against *E. faecium* and synergistic activity at higher concentrations against *E. faecalis*, in line with previous studies reporting enhanced phage access through daptomycin-induced membrane disruption (15, 38). Fosfomycin-phage treatment overcame the limited efficacy of fosfomycin monotherapy, showing synergy at 1× MIC and complete eradication at higher concentrations likely due to phage-mediated disruption of the biofilm matrix, enhancing fosfomycin penetration into bacterial cells, as supported by prior work (39). Similarly, further research demonstrated that combining phages with fosfomycin enhanced bactericidal activity, reduced the emergence of resistant bacterial clones, and decreased the MIC values compared to fosfomycin monotherapy (40, 41). Together, these findings highlight that phage-antibiotic synergy in enterococci may result from reduced resistance emergence, lowered antibiotic MICs, and improved biofilm penetration. This variability further suggests that tailoring phage selection to individual strains may maximize therapeutic success (36).

This study has some limitations. First, only two strains (a clinical *E. faecium* VRE isolate and *E. faecalis* ATCC 51299) were available for phage characterization, as lytic phages could not be recovered for the other screened isolates. Although this restricts representativeness, the findings provide proof of concept for phage-antibiotic synergy in biofilm-associated infections. Future work should include a broader strain panel to capture the genetic and phenotypic diversity of VRE. Second, antibiofilm activity was assessed only by isothermal microcalorimetry. This method provides sensitive, real-time, and non-invasive monitoring of metabolic activity, but it does not directly quantify bacterial numbers and requires cautious interpretation. Still, previous studies demonstrate that parameters, such as time to detection and peak heat flow, correlate well with CFU counts, supporting its validity (42).

In conclusion, this study demonstrates the potential of phage-antibiotic combinations, particularly involving dalbavancin and fosfomycin, as a powerful strategy against *Enterococcus* spp. biofilms. The novel lytic phages CUB-FM and CUB-FS showed strain-specific efficacy and genomic safety profiles, supporting their therapeutic use. Importantly, the synergistic interactions observed in our *in vitro* models may offer a pathway to improved clinical outcomes in complex biofilm-associated infections, including those caused by vancomycin-resistant enterococci. Further *in vivo* studies and clinical trials are warranted to explore optimal dosing regimens and to validate these promising *in vitro* findings.

## MATERIALS AND METHODS

### Bacterial strains and bacteriophages

Initially, four clinical *Enterococcus faecium* VRE isolates, one clinical *Enterococcus gallinarum* VRE isolate, and the reference *Enterococcus faecalis* VRE ATCC 51299 strain were included for screening. However, lytic phages were recovered only against one clinical *E. faecium* isolate and *E. faecalis* ATCC 51299, which were, therefore, selected for further characterization. Clinical isolates of *Enterococcus* spp. obtained from a bone and joint infection site (Labor Berlin—Charité Vivantes GmbH, Berlin, Germany) and the laboratory reference strain *E. faecalis* ATCC 51299 were stored at −80°C using a cryovial bead preservation system (Microbank, Pro-Lab Diagnostics, Canada). For bacterial strains, DNA was extracted using the DNeasy UltraClean Microbial Kit (Qiagen) and prepared for whole-genome sequencing on an Illumina Miniseq device as previously described (43). An assembly was constructed using Unicycler (Galaxy v0.5.1) (44). ResFinder (v4.7.2; DTU Center for Genomic Epidemiology; minimum DNA identity of 90% and minimum coverage of 60%) was used to determine resistance genes present in the isolates (45). Virulence factors and biofilm-associated genes were identified using the same thresholds from the VFDB database (46). PHASTEST v3.0 (47) was run for prophage detection in the genomes.

Strictly lytic bacteriophages targeting *E. faecium* (CUB-FM phage) and *E. faecalis* (CUB-FS phage) were isolated from hospital sewage and subsequently employed in experimental procedures. The phage isolation procedure employed an enrichment method detailed in a previous study (48). To ensure phage purity, four consecutive cycles of single-plaque isolation were conducted on the bacterial host strain. For propagation, the host strain was grown overnight at 37°C in tryptic soy broth (TSB; US Biological, Eching, Germany). Next, 1 mL of phage lysate at a concentration of approximately $10^8$ PFU/mL was incubated at 4°C for 1 h. An aliquot of 0.2 mL overnight culture was inoculated into 20 mL TSB and incubated with shaking at 37°C until $OD_{600}$ reached 0.4, after which 0.1 mL of the phage suspension was added. The culture was incubated until complete clearing (~5 h or overnight if necessary). The lysate was centrifuged at 4,000 × $g$ for 20 min, and the supernatant was sterilized using 0.22 µm filter. Polyethylene glycol (PEG) precipitation was carried out by adding PEG/NaCl stock solution (20% w/v PEG-8000, 2.5 M NaCl; PanReac AppliChem, Darmstadt, Germany) to the lysate at 25% of its volume (e.g., 250 µL per 1 mL lysate), mixing by inversion, and incubating on ice for ≥1 h. Phages were pelleted by centrifugation at 13,000 × $g$ (4°C, 40 min), the supernatant discarded, and residual liquid removed. Pellets were gently resuspended in 100 µL SM buffer (10 mM Tris-HCl, pH 7.5, 100 mM NaCl, 8 mM $MgSO_4$) and incubated at room temperature for 1 h with occasional pipetting. Final titers were determined by plaque assay on the host strain by serial dilution. Subsequently, the phage was cultured in a liquid medium, with bacterial strains grown in tryptic soy broth (TSB) (US Biological, Eching, Germany) overnight at 37°C. These phage stocks were used for whole-genome sequencing analysis as detailed in Sharifi et al. (49). The phage's taxonomy was determined using Virus Intergenomic Distance Calculator (VIRIDIC) v1.1 analysis (50). The phage genomes of CUB-FM and CUB-FS were submitted to NCBI GenBank and are available under accession numbers PV646496 and PV646497, respectively.

### Antibiotics

Dilution of antibiotics was prepared according to both Clinical and Laboratory Standards Institute (CLSI) (51) and European Committee on Antimicrobial Susceptibility Testing (EUCAST) (52). Fosfomycin was dissolved in sterile distilled water with glucose-6-phosphate (500 µL glucose-6-phosphate per 10 mL of Muller Hinton Broth [MHB]), daptomycin in sterile distilled water with calcium chloride, and dalbavancin in dimethyl sulfoxide DMSO) with polysorbate-80, immediately prior to utilization.

## Morphological analysis of the phages by TEM

The morphology of the phages was analyzed by TEM following negative staining, as previously described (53). Briefly, 15 µL of phage suspension was placed onto parafilm and then transferred onto a carbon-coated, glow-discharged (Leica Microsystems, Wetzlar, Germany) Ni-mesh grid (G2430N; Plano GmbH, Wetzlar, Germany). The phage particles were allowed to adsorb for 10–15 min at room temperature. Subsequently, the grids were washed three times with distilled water and negatively stained with 1% aqueous uranyl acetate (SERVA Electrophoresis GmbH, Heidelberg, Germany) for 20 s. After air drying, the grids were examined with a Zeiss EM 906 transmission electron microscope (Carl Zeiss Microscopy Deutschland GmbH, Oberkochen, Germany) at a voltage of 80 kV. Phage dimensions were determined with the image processing software ImageJ.JS (54, 55).

## MIC determination of antibiotics and phage efficacy testing

Minimal inhibitory concentrations (MICs) for the antibiotics were obtained by the E-test (Liofilchem, Roseto degli Abruzzi, Italy) on Muller Hinton agar based on CLSI breakpoints and EUCAST ECOFFs (51, 52).

To determine phage efficacy, biofilms of *E. faecium* as well as *E. faecalis* were formed on sterile porous glass beads (diameter 4 mm, pore sizes 60 µm, ROBU1, Hattert, Germany) (56), washed three times with 2 mL 0.9% NaCl to remove planktonic bacteria, and subsequently exposed to the phage dilutions ranging from $10^5$ to $10^{13}$ PFU/mL. The effects of phage treatment were monitored over a 24 h period using isothermal microcalorimetry.

## Antibiotics alone and phage-antibiotic combinations against biofilm

After biofilm formation as described above, beads were transferred to 24-well plates containing 1 mL of MHB with different concentrations of the antibiotic (1, 10, and 100× MIC) or phage (at defined PFU/mL) as monotherapy or phage-antibiotic combinations. For all phage-antibiotic combinations, biofilms were initially treated with 500 µL of phage solution in MHB at twice the intended final concentration, ensuring that after the subsequent addition of the antibiotic, the concentration of both agents would be at the desired level. The phage treatment was carried out for 4 h at 37°C. Following this, 500 µL of the corresponding antibiotic solution also prepared at twice the desired final concentration was added, and the biofilms were incubated for an additional 20 h at 37°C. After a total incubation of 24 h, treated biofilm beads were rinsed three times with 0.9% NaCl, placed in sterile glass ampules with 3 mL fresh TSB, and inserted in the calorimeter, where heat produced by viable bacteria present in the bead after 24 h of treatment or no treatment (growth control) was monitored for 36 h.

## Microcalorimetry setup and heat measurement assay

Biofilm treatment efficacy was assessed using an isothermal calorimeter TAM III (TA Instruments, New Castle, DE, USA), as previously described (57, 58). This technique allows real-time monitoring of bacterial metabolic activity by measuring heat flow as an indirect indicator of microbial replication. To evaluate the antimicrobial effects of the treatments, microcalorimetric data were collected by recording the heat generated from the regrowth of viable bacteria following 24 h of exposure to antibiotics, phages, or their combinations. The assay measured two primary parameters: (i) maximum heat flow peaks ($HF_{Max}$, µW) to assess metabolic activity; and (ii) time to reach maximum heat ($t_{Max}$, h) to evaluate delays in bacterial regrowth. To facilitate comparison between treatments, results were expressed as $\Delta t_{Max}$ and $\Delta HF_{Max}$ relative to the untreated growth control for each strain and experiment:

$\Delta t_{Max} = t_{Max}treatment - t_{Max}control$
$\Delta HF_{Max} = HF_{Max}treatment - HF_{Max}control$

Positive $\Delta t_{Max}$ values indicate delayed bacterial growth relative to the control, while negative $\Delta HF_{Max}$ values indicate reduced metabolic activity. Measurements were recorded every 120 s over a 36 h period. A dose-dependent antimicrobial effect was indicated by a reduction in $HF_{Max}$ values and a delay in $t_{1Max}$ compared to the control, in line with previous methodologies (57, 59). The synergy between antibiotics and phages was defined as a combined effect exceeding the sum of individual effects, whereas additive effects represent improved activity over monotherapy but within the expected sum of the individual effects. All experiments were performed in triplicate to ensure reproducibility.

## Statistical analysis

Results were expressed as mean and standard deviation for normally distributed variables (further analyzed by Student $t$ test) or median and minimum/maximum for non-normally distributed variables (analyzed by Mann-Whitney U test). Hypothesis testing was two-tailed, with $P < 0.05$ considered statistically significant. Software GraphPad Prism v9.3 (GraphPad Software Inc., La Jolla, CA, USA) was used for statistical analysis and for graphics.

## ACKNOWLEDGMENTS

We thank the Berliner Wasserbetriebe for their support in the collection of hospital sewage. The authors also thank the Core Facility for Electron Microscopy at the Charité—Universitätsmedizin Berlin for the help in the collection of transmission electron microscopy images.

This work was supported by an educational grant of the PRO-IMPLANT Foundation, Berlin, Germany (https://www.pro-implant.org), a non-profit organization supporting research, education, global networking, and care of patients with bone, joint, or implant-associated infection. A.I.-V.-S. received support for the study through the European Society of Clinical Microbiology and Infectious Diseases (ESCMID) Mentorship Programme, as well as a training grant from the Spanish Society of Clinical Microbiology and Infectious Diseases (SEIMC).

## AUTHOR AFFILIATIONS

[1]Center for Musculoskeletal Surgery (CMSC), Charité—Universitätsmedizin Berlin, Corporate Member of Freie Universität Berlin, Humboldt-Universität zu Berlin, Berlin Institute of Health, Berlin, Germany
[2]Clinical Microbiology and Infectious Diseases Department, Hospital General Universitario Gregorio Marañón, Madrid, Spain
[3]Department of Biosystems, KU Leuven, Leuven, Belgium
[4]School of Medicine, Faculty of Health, Queensland University of Technology, Brisbane, Queensland, Australia
[5]Royal Brisbane and Women's Hospital, Herston, Queensland, Australia

## AUTHOR ORCIDs

Alvaro Irigoyen-von-Sierakowski ⓘ http://orcid.org/0000-0002-5137-8818
Jeroen Wagemans ⓘ http://orcid.org/0000-0002-2185-5724
Yu Ning ⓘ http://orcid.org/0009-0006-2517-8907
Andrej Trampuz ⓘ http://orcid.org/0000-0002-5219-2521
Svetlana Karbysheva ⓘ http://orcid.org/0000-0002-9265-5260

## AUTHOR CONTRIBUTIONS

Rima Fanaei Pirlar, Formal analysis, Investigation, Methodology, Supervision, Validation, Visualization | Alvaro Irigoyen-von-Sierakowski, Formal analysis, Investigation, Methodology, Supervision, Validation, Visualization | Jeroen Wagemans, Formal analysis, Investigation, Methodology, Validation, Visualization | Yu Ning, Investigation | Rob Lavigne,

Resources | Andrej Trampuz, Conceptualization, Funding acquisition, Project administration, Resources, Supervision, Visualization, Writing – review and editing | Svetlana Karbysheva, Conceptualization, Data curation, Formal analysis, Investigation, Methodology, Project administration, Resources, Supervision, Validation, Visualization

## DATA AVAILABILITY

Bacteriophage whole genome sequence data presented in this study are openly available in the NCBI GenBank under accession numbers PV646496 and PV646497.

## ADDITIONAL FILES

The following material is available online.

### Supplemental Material

**Supplemental material (Spectrum01818-25-s0001.pdf).** Table S1; Fig. S1 and S2.

### Open Peer Review

**PEER REVIEW HISTORY (review-history.pdf).** An accounting of the reviewer comments and feedback.

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
