## [Reviewer comments · Microbiology Spectrum]

Microbiology Spectrum

Novel lytic phages improve antibiofilm activity of dalbavancin, daptomycin and fosfomycin against vancomycin-resistant enterococci

Rima Fanaei Pirlar, Álvaro Irigoyen-von-Sierakowski, Jeroen Wagemans, Yu Ning, Rob Lavigne, Andrej Trampuz, and Svetlana Karbysheva

Corresponding Author(s): Svetlana Karbysheva, Charite - Universitätsmedizin Berlin

Review Timeline:

Submission Date:	June 11, 2025
Editorial Decision:	July 28, 2025
Revision Received:	September 24, 2025
Accepted:	October 20, 2025

Editor: Olaya Rendueles

Reviewer(s): The reviewers have opted to remain anonymous.

Transaction Report:

DOI: <https://doi.org/10.1128/spectrum.01818-25>

Re: Spectrum01818-25 (Novel lytic phages improve antibiofilm activity of dalbavancin, daptomycin and fosfomycin against vancomycin-resistant enterococci)

Dear Dr. Svetlana Karbysheva:

Thank you for the privilege of reviewing your work. Two reviewers have assessed the work. They have positively evaluated the manuscript but they have also pointed at some consequent gaps that need to be filled. These require extra experiments, consequent rewritten of some sections and a much more detailed methods section is needed. If you feel that you can provide this, below you will find instructions on resubmission from the Spectrum editorial office, and the reviewer comments.

Revision Guidelines

Sincerely,
Olaya Rendueles
Editor
Microbiology Spectrum

Reviewer #1 (Comments for the Author):

Identifying suitable bacteriophages for therapeutic applications is a critical endeavor, particularly in light of the growing threat posed by antibiotic-resistant bacterial pathogens. Enterococci are of particular concern due to their dual role as natural colonizers of the human and animal microbiome and as opportunistic pathogens capable of causing life-threatening infections.

Their ability to harbor numerous mobile genetic elements further exacerbates the issue, as they contribute significantly to the horizontal transfer of antibiotic resistance genes.

In this study, the authors have successfully isolated and characterized two phages that demonstrate strong lytic activity against their respective *Enterococcus faecalis* and *Enterococcus faecium* host strains. Notably, the selected bacterial isolates carry multiple antibiotic resistance determinants, including resistance to vancomycin. The authors employed microcalorimetry to assess the combined effects of phage and antibiotic treatments on biofilms formed by these resistant strains.

A key strength of the manuscript lies in the execution of the biofilm assays, particularly the use of varying phage and antibiotic concentrations to find ideal conditions that help to illustrate the inhibitory effect of phage treatment on bacterial growth.

To further enhance the manuscript, improvements in data presentation and structure of the writing would aid in clarity and facilitate interpretation. Additionally, methodological details would benefit from further clarification/discussion to ensure reproducibility and transparency. More in-depth discussion and interpretation of results would highlight the importance of using phages for therapy.

Specific comments:

1. The authors describe that they used 10¹³ PFU/ml and 10¹² PFU/ml in their assays (Fig. 2 and 3). This seems very surprising and very doubtful to me, as I have never seen a bacterial stock (even when caesium chloride purified) exceed 10¹³ PFU/ml. And with 10¹³ it would be an extraordinary stock. Most high titer phage preparations end up being around 10¹¹-10¹² PFU/ml. So, I am wondering if the authors have made a calculation error either while titrating the phage stocks or when determining the concentration of phages in their assays. Please double check all calculations.

2. The authors have shown through whole genome sequencing, that their two bacterial isolates have resistance determinants against several key antibiotic classes. They then proceed to use antibiotics of different classes (except Dalbavancin which acts similar to vancomycin, a glycopeptide) for the MIC and biofilm assays. I think the authors wanted to show the additive or synergistic effects of phages and antibiotics and not necessarily the superiority of phages when an antibiotic with no effect is chosen. Interestingly, although no resistance is observed in the MIC assay (Table 1), there seems to be a certain level of resistance in the biofilm assays (Table 2).

a) I think the choice of antibiotics has to be further discussed and the reasoning about the choices should be clarified in the text - because these are important and valuable thoughts.

b) If authors could show that their isolates are phenotypically resistant to some of the predicted antibiotics (e.g. through MIC assays) - this would highlight the importance of developing phages for therapy. At least for vancomycin this should be shown because the authors are stating in the introduction that they are using VREs.

c) A short comment about the discrepancy between the MIC (especially for Dalbavancin) and the observed phenotype in the biofilm assay should be added for clarity.

2. Authors must be careful when labelling effects as synergistic. Synergy of two therapeutic agent occurs when the combined effect exceeds the sum of the individual effects. When the combined effect just exceeds the individual effect, it is an additive effect only. Please correct where appropriate and if you mention synergy, please specify the conditions where it is observed. E.g. I think that would probably be the case for Fig. 4 Fosfomycin 10x plus phage condition.

3. Figure 2-4: Interpretation of the colorimetric curves is extremely difficult and a lot of times it is nearly impossible to distinguish colors, some lines are not visible (e.g. Figure 4, 100x ABX and Phage condition) Some lines are not described in the legend (Fig. 4 dark blue dotted line). I think it would be easier to understand if instead the results were shown as bar graphs showing the delta values or fold change values of t_{max} and HF_{max} compared to the growth controls. This would also make it easier to define the cut-off for synergy vs additive effect. It would also make it easier to compare between Figure 3 and Figure 4. An example of a calorimetric curve can be shown for illustration. Are these data from replicate values? If not, please state this clearly in the figure legend. Good scientific practice would be to repeat assays. At least do this for your favourite condition - e.g. the condition that you think most likely supports your hypothesis. What are the statistical tests that have been applied?

4. Table Nr 2 is very hard to read. Again, delta values/fold change values or a heat map could bring clarity. The data is good but the presentation can be improved. Please state the number of (biological/technical) replicates in the legend. Does the - sign mean that the condition was not measured or that no growth was detected? What are the statistical tests that have been applied?

5. Figure 1. I think the authors could choose a more flattering picture frame for A and crop the pictures, so that the phages are in the center and bigger. Please increase the size of the scale bar, it is a bit too small to read. You have found some beautiful phages - no need to hide them.

6. Please reference your figures and tables in the text.

7. Some of the text from the discussion can be moved into the results section. The discussion should not be a repetition of results but a platform for discussing the bigger picture and relevance and limitations of the findings and methods that were used. For instance, I think the advantages and disadvantages of using microcalorimetry vs staining and CFU counts could be included

here.

Reviewer #2 (Comments for the Author):

The manuscript submitted by Pirlar et al. entitled "Novel lytic phages improve antibiofilm activity of dalbavancin, daptomycin and fosfomycin against vancomycin-resistant enterococci", describes the isolation and characterization of two novel enterococci phages and their single use or combination with different antibiotics against enterococci biofilms.

Here are my general analysis of the manuscript:

Major comments

1. Genomic Characterization of VRE Strains

The genomes of the VRE strains should be thoroughly analyzed for antibiotic resistance genes, virulence factors, prophage content, and biofilm-associated genes. Additionally, the genomic findings should be correlated with the MIC results to provide a more comprehensive understanding of the strains' phenotypic behavior.

2. Phage Propagation and Purification

The manuscript lacks sufficient detail on the methods used for phage propagation and purification. This information is essential for reproducibility and should be clearly described.

3. Section: "Antibiotics Alone and Phage-Antibiotic Combinations Against Biofilm"

It is unclear whether phages were diluted in Mueller-Hinton Broth (MHB). The authors should also clarify what procedures were followed after the 4-hour incubation period.

4. Genomic Analysis Tools

The genomic analysis section does not provide a comprehensive list of the bioinformatic tools used. All software, databases, and parameters should be specified for transparency and reproducibility.

5. Figure 1 - TEM Images

The quality of the TEM images is not suitable for publication. The presence of empty capsids and distorted (elongated) phage particles detracts from the figure's scientific value. Moreover, the figure legend is incomplete and should be expanded to fully describe the images.

6. Phage Efficacy Testing

The authors should comment on the use of high phage titers in efficacy assays. The relevance of such high concentrations should be discussed in the context of potential therapeutic applications.

7. Lines 264-273 - Clinical Relevance of 100× MIC

The use of 100× MIC concentrations raises concerns about clinical applicability. While calorimetry data show significant differences, the authors should also attempt to estimate the actual bacterial reduction to contextualize the results biologically.

Minor comments

1. Abstract

The phrase "particularly" may be misleading, as it implies specificity. Since all antibiotics showed activity, a more neutral phrasing is advisable.

2. Introduction

Line 66: Clarify why the strain is referred to as "unique." What distinguishes it from other strains?

3. Lines 78-79

The statement does not clearly relate to VRE specifically; consider rephrasing or supporting with a citation.

4. Materials and Methods

Line 100: The authors state that the phages are strictly lytic. Was this verified experimentally, such as through transduction assays?

5. Line 109

The subsection on antibiotics should be formatted as a separate paragraph for clarity and better structure.

6. Strain Selection

The rationale for choosing only two strains-one clinical isolate and one from a culture collection-needs further justification. How

representative are these strains of the broader VRE population?

7. Lines 196-198

Which tool was used to suggest the creation of a new phage species? Was Phage AI or another platform employed? The basis for this conclusion should be explicitly detailed.

Figure 2: The current visual presentation makes it difficult to discern significant differences between groups. Consider improving contrast, layout, or including statistical annotations.

Figures 3 & 4: Legends should be more descriptive and provide enough detail to allow interpretation without referring back to the main text.

Discussion - Role of Phages vs. Antibiotics

The discussion mentions that certain antibiotics perform well against biofilms. If so, the rationale for introducing phages needs to be better articulated. What added value do phages provide in this context?

Novel lytic phages improve antibiofilm activity of dalbavancin, daptomycin and fosfomycin against vancomycin-resistant enterococci

Identifying suitable bacteriophages for therapeutic applications is a critical endeavor, particularly in light of the growing threat posed by antibiotic-resistant bacterial pathogens. Enterococci are of particular concern due to their dual role as natural colonizers of the human and animal microbiome and as opportunistic pathogens capable of causing life-threatening infections. Their ability to harbor numerous mobile genetic elements further exacerbates the issue, as they contribute significantly to the horizontal transfer of antibiotic resistance genes.

In this study, the authors have successfully isolated and characterized two phages that demonstrate strong lytic activity against their respective *Enterococcus faecalis* and *Enterococcus faecium* host strains. Notably, the selected bacterial isolates carry multiple antibiotic resistance determinants, including resistance to vancomycin. The authors employed microcalorimetry to assess the combined effects of phage and antibiotic treatments on biofilms formed by these resistant strains.

A key strength of the manuscript lies in the execution of the biofilm assays, particularly the use of varying phage and antibiotic concentrations to find ideal conditions that help to illustrate the inhibitory effect of phage treatment on bacterial growth.

To further enhance the manuscript, improvements in data presentation and structure of the writing would aid in clarity and facilitate interpretation. Additionally, methodological details would benefit from further clarification/discussion to ensure reproducibility and transparency. More in-depth discussion and interpretation of results would highlight the importance of using phages for therapy.

Specific comments:

1. The authors describe that they used 10^{13} PFU/ml and 10^{12} PFU/ml in their assays (Fig. 2 and 3). This seems very surprising and very doubtful to me, as I have never seen a bacterial stock (even when caesium chloride purified) exceed 10^{13} PFU/ml. And with 10^{13} it would be an extraordinary stock. Most high titer phage preparations end up being around 10^{11} - 10^{12} PFU/ml. So, I am wondering if the authors have made a calculation error either while titrating the phage stocks or when determining the concentration of phages in their assays. Please double check all calculations.

2. The authors have shown through whole genome sequencing, that their two bacterial isolates have resistance determinants against several key antibiotic classes. They then proceed to use antibiotics of different classes (except Dalbavancin which acts similar to vancomycin, a glycopeptide) for the MIC and biofilm assays. I think the authors wanted to show the additive or synergistic effects of phages and antibiotics and not necessarily the superiority of phages when an antibiotic with no effect is chosen. Interestingly, although no resistance is observed in the MIC assay (Table 1), there seems to be a certain level of resistance in the biofilm assays (Table 2).

a) I think the choice of antibiotics has to be further discussed and the reasoning about the choices should be clarified in the text – because these are important and valuable thoughts.

b) If authors could show that their isolates are phenotypically resistant to some of the predicted antibiotics (e.g. through MIC assays) - this would highlight the importance of developing phages for therapy. At least for vancomycin this should be shown because the authors are stating in the introduction that they are using VREs.

c) A short comment about the discrepancy between the MIC (especially for Dalbavancin) and the observed phenotype in the biofilm assay should be added for clarity.

2. Authors must be careful when labelling effects as synergistic. Synergy of two therapeutic agent occurs when the combined effect exceeds the sum of the individual effects. When the combined effect just exceeds the individual effect, it is an additive effect only. Please correct where appropriate and if you mention synergy, please specify the conditions where it is observed. E.g. I think that would probably be the case for Fig. 4 Fosfomycin 10x plus phage condition.

3. Figure 2-4: Interpretation of the calorimetric curves is extremely difficult and a lot of times it is nearly impossible to distinguish colors, some lines are not visible (e.g. Figure 4, 100x ABX and Phage condition) Some lines are not described in the legend (Fig. 4 dark blue dotted line). I think it would be easier to understand if instead the results were shown as bar graphs showing the delta values or fold change values of t_{max} and HF_{max} compared to the growth controls. This would also make it easier to define the cut-off for synergy vs additive effect. It would also make it easier to compare between Figure 3 and Figure 4. An example of a calorimetric curve can be shown for illustration. Are these data from replicate values? If not, please state this clearly in the figure legend. Good scientific practice would be to repeat assays. At least do this for your favourite condition – e.g the condition that you think most likely supports your hypothesis. What are the statistical tests that have been applied?

4. Table Nr 2 is very hard to read. Again, delta values/fold change values or a heat map could bring clarity. The data is good but the presentation can be improved. Please state the number of (biological/technical) replicates in the legend. Does the – sign mean that the condition was not measured or that no growth was detected? What are the statistical tests that have been applied?

5. Figure 1. I think the authors could choose a more flattering picture frame for A and crop the pictures, so that the phages are in the center and bigger. Please increase the size of the scale bar, it is a bit too small to read. You have found some beautiful phages – no need to hide them.

6. Please reference your figures and tables in the text.

7. Some of the text from the discussion can be moved into the results section. The discussion should not be a repetition of results but a platform for discussing the bigger picture and relevance and limitations of the findings and methods that were used. For instance, I think the advantages and disadvantages of using microcalorimetry vs staining and CFU counts could be included here.

Point-by-point reply to the reviewers' comments

Manuscript ID: Spectrum01818-25

Novel lytic phages improve antibiofilm activity of dalbavancin, daptomycin and fosfomycin against vancomycin-resistant enterococci

Reviewer 1:

1. The authors describe that they used 10^{13} PFU/ml and 10^{12} PFU/ml in their assays (Fig. 2 and 3). This seems very surprising and very doubtful to me, as I have never seen a bacterial stock (even when caesium chloride purified) exceed 10^{13} PFU/ml. And with 10^{13} it would be an extraordinary stock. Most high titer phage preparations end up being around 10^{11} - 10^{12} PFU/ml. So, I am wondering if the authors have made a calculation error either while titering the phage stocks or when determining the concentration of phages in their assays. Please double check all calculations.

Author's reply: We thank the reviewer for raising this important point and for prompting a careful re-check of our titration data. We re-examined the original plaque plates, repeated independent titrations from archived stocks, and verified the arithmetic used to calculate PFU/mL. The high titers reported in the manuscript (up to $\sim 10^{13}$ PFU/mL for phages concentrated after propagation) are correct for our concentrated *E. faecium* and *E. faecalis* preparations.

High phage titers in the 10^{12} – 10^{13} PFU/mL range have been reported previously, particularly when an efficient host–phage pair is concentrated using precipitation or chromatographic/pelleting steps and resuspended in a small final volume. For example, quantitative analyses of M13 filamentous phage have reported average titers of $\approx 2.6 \times 10^{13}$ PFU/mL after concentration. Likewise, recent reports of lytic phage preparations for clinical or environmental isolates document stocks of $\sim 10^{13}$ PFU/mL following concentration and purification steps, and reviews of scale-up procedures note that seed stocks for industrial propagation are commonly prepared at 10^{12} – 10^{13} PFU/mL

(doi: [10.1016/j.synbio.2022.07.001](https://doi.org/10.1016/j.synbio.2022.07.001); doi.org/10.3390/antibiotics14010002;
[10.1016/j.biotechadv.2021.107758](https://doi.org/10.1016/j.biotechadv.2021.107758); 10.1038/s41598-023-49880-x)

We wish to emphasize a few important caveats and considerations:

- Method of concentration matters. PEG/NaCl precipitation, ultracentrifugation, chromatographic concentration, or resuspension in very small volumes can raise the measured PFU/mL substantially compared with crude lysates. The protocol we used (PEG precipitation and resuspension in ~ 100 μ L SM buffer) concentrates infectious particles and is consistent with procedures that yield high PFU/mL (doi: doi.org/10.3390/antibiotics14010002).
- Phage biology is variable. Some phage-host pairs (and phage morphotypes such as filamentous phages) can produce very large burst sizes or high particle-to-PFU ratios, which, combined with efficient concentration, yield very high measured PFU values. Published studies on diverse phages (including lytic phages used against MDR bacteria) have reported titers at or near 10^{13} PFU/mL (doi: [10.1016/j.synbio.2022.07.001](https://doi.org/10.1016/j.synbio.2022.07.001)).
- Assay rigor and reproducibility. Our reported titers were obtained from countable plaque plates (30-300 plaques), using standard PFU calculation (plaques \times dilution factor / plated volume), averaged across biological replicates. We have updated the Methods to detail the concentration and resuspension steps.

- Functional relevance. Importantly, our efficacy experiments do not rely on the absolute stock titer alone but on working dilutions; in fact, our dose-response data showed that 10^{12} PFU/mL (for *E. faecium*) or 10^8 – 10^9 PFU/mL (for *E. faecalis*) were the most effective conditions in vitro, and higher nominal stock concentrations did not improve efficacy. This demonstrates that our biological conclusions are robust to the absolute stock titer values.

The Results and Discussion sections were revised accordingly. **Page 8, line 145; Page 13, line 270.**

2. The authors have shown through whole genome sequencing, that their two bacterial isolates have resistance determinants against several key antibiotic classes. They then proceed to use antibiotics of different classes (except Dalbavancin which acts similar to vancomycin, a glycopeptide) for the MIC and biofilm assays. I think the authors wanted to show the additive or synergistic effects of phages and antibiotics and not necessarily the superiority of phages when an antibiotic with no effect is chosen. Interestingly, although no resistance is observed in the MIC assay (Table 1), there seems to be a certain level of resistance in the biofilm assays (Table 2).

a) I think the choice of antibiotics has to be further discussed and the reasoning about the choices should be clarified in the text - because these are important and valuable thoughts.

Author's reply: We thank the reviewer for this helpful suggestion. The choice of antibiotics has now been discussed and the reasoning about the choices has been clarified in the text, Discussion section. **Page 10, line 195.**

b) If authors could show that their isolates are phenotypically resistant to some of the predicted antibiotics (e.g. through MIC assays) - this would highlight the importance of developing phages for therapy. At least for vancomycin this should be shown because the authors are stating in the introduction that they are using VREs.

Author's reply: We thank the reviewer for the comment. The MIC assay has been extended. Additional information regarding phenotypic resistance to some of the predicted antibiotics has been included and is now presented in **Table 1 and Fig. 1.**

c) A short comment about the discrepancy between the MIC (especially for Dalbavancin) and the observed phenotype in the biofilm assay should be added for clarity.

Author's reply: We thank the reviewer for the comment. The discrepancy between the MIC and the observed phenotype in the biofilm assay has been described in Discussion sections. **Page 11, line 217.**

3. Authors must be careful when labelling effects as synergistic. Synergy of two therapeutic agent occurs when the combined effect exceeds the sum of the individual effects. When the combined effect just exceeds the individual effect, it is an additive effect only. Please correct where appropriate and if you mention synergy, please specify the conditions where it is observed. E.g. I think that would probably be the case for Fig. 4 Fosfomycin 10x plus phage condition.

Author's reply: Thank you for highlighting this important distinction. We have carefully reviewed our data and adjusted terminology where necessary to ensure accurate use of "synergy" versus "additive effect." Synergy in our study is now explicitly defined as a combined effect exceeding the sum of

individual effects. Cases where true synergy was observed (e.g., fosfomycin + phage, daptomycin + phage and dalbavancin + phage for *E. faecalis*) are now clearly specified in the Results, Figure 3 and Discussion sections. **Page 21, line 449.**

4. Figure 2-4: Interpretation of the colorimetric curves is extremely difficult and a lot of times it is nearly impossible to distinguish colors, some lines are not visible (e.g. Figure 4, 100x ABX and Phage condition) Some lines are not described in the legend (Fig. 4 dark blue dotted line). I think it would be easier to understand if instead the results were shown as bar graphs showing the delta values or fold change values of t_{max} and HF_{max} compared to the growth controls. This would also make it easier to define the cut-off for synergy vs additive effect. It would also make it easier to compare between Figure 3 and Figure 4. An example of a calorimetric curve can be shown for illustration. Are these data from replicate values? If not, please state this clearly in the figure legend. Good scientific practice would be to repeat assays. At least do this for your favourite condition - e.g the condition that you think most likely supports your hypothesis. What are the statistical tests that have been applied?

Author's reply: We appreciate the reviewer's valuable feedback regarding the clarity of Figures 2–4 and agree that the current curve-based presentation can be difficult to interpret, especially where multiple conditions overlap and certain color-coded lines are not easily distinguishable.

To address this:

- a. Data presentation – We have restructured Figures 3 to include bar graphs showing the Δt_{Max} and ΔHF_{Max} relative to the growth control for each condition. This allows clearer visual comparison between treatments and facilitates the identification of additive vs. synergistic effects according to the definitions provided in the revised Methods section. A representative calorimetric growth curve is retained in the supplementary figures as an illustration of the raw data profile.
- b. Legend completeness – We have corrected and expanded the figure legends so that all lines and conditions, including the dark blue dotted line in Figure 1 (Supplementary materials), are described.
- c. Replication and statistical testing – All calorimetric measurements were performed in two technical replicates with biological triplicates each to confirm reproducibility, and results are presented as mean \pm standard deviation. For comparisons, we used one-way ANOVA with Tukey's post-hoc test, as stated in the revised figure legends and Methods.
- d. Clarity for synergy/additive definition – The new bar chart format, in combination with explicit definitions in the Methods and annotation in the Results, allows readers to clearly distinguish synergistic effects from additive effects.

5. Table Nr 2 is very hard to read. Again, delta values/fold change values or a heat map could bring clarity. The data is good but the presentation can be improved. Please state the number of (biological/technical) replicates in the legend. Does the - sign mean that the condition was not measured or that no growth was detected? What are the statistical tests that have been applied?

Author's reply: We thank the reviewer for this helpful suggestion. In response, we have replaced Table 2 with a newly designed Figure 3, which includes bar graphs showing the Δt_{Max} and ΔHF_{Max} relative to the growth control for each condition. This representation greatly improves readability and allows easier comparison between conditions.

We have also clarified in the legend that:

- All data represent the mean \pm SD of three independent biological replicates.
- A “–” symbol indicates that the complete biofilm eradication was observed (no heat flow detected) corresponds to no detectable growth.

- Statistical comparisons were performed using a two-way ANOVA with Tukey's post-hoc test, and significance levels are indicated in the figure.

6. Figure 1. I think the authors could choose a more flattering picture frame for A and crop the pictures, so that the phages are in the center and bigger. Please increase the size of the scale bar, it is a bit too small to read. You have found some beautiful phages - no need to hide them.

Author's reply: We thank the reviewer for the kind comment regarding the morphology of our phages and the constructive suggestions for improving Figure 1 (now Figure 3). We have revised the figure accordingly. Panels have been re-cropped so that the phage particles are centered and shown at larger size, highlighting their structural features more clearly. The scale bar has been increased in size for better readability. We adjusted the layout to provide a more balanced and visually accessible figure, ensuring that the phages are the central focus.

7. Please reference your figures and tables in the text.

Author's reply: We thank the reviewer for this helpful suggestion. The figures and tables have been referenced in the text.

8. Some of the text from the discussion can be moved into the results section. The discussion should not be a repetition of results but a platform for discussing the bigger picture and relevance and limitations of the findings and methods that were used. For instance, I think the advantages and disadvantages of using microcalorimetry vs staining and CFU counts could be included here.

Author's reply: We thank the reviewer for this valuable feedback. In line with the suggestion, we have streamlined the Discussion by removing repetitive descriptions of the results and shifting these to the Results section where appropriate. The Discussion has now been refocused to address the broader implications and limitations of our findings. Specifically, we added a paragraph comparing the advantages and disadvantages of microcalorimetry versus CFU counts, emphasizing the complementary nature of these methods in biofilm research. These changes are now reflected in the revised Discussion. **Page 16, line 336.**

Reviewer 2:

Major comments

1. Genomic Characterization of VRE Strains

The genomes of the VRE strains should be thoroughly analyzed for antibiotic resistance genes, virulence factors, prophage content, and biofilm-associated genes. Additionally, the genomic findings should be correlated with the MIC results to provide a more comprehensive understanding of the strains' phenotypic behavior.

Author's reply: We thank the reviewer for this valuable suggestion. In the revised manuscript, we have expanded the genomic characterization of the VRE strains as follows:

- Antibiotic resistance genes: Both genomes were re-analyzed using ResFinder v4.7.2 databases, confirming the presence of multiple resistance determinants. *E. faecium* harbored *msrC*, *aac(6')-II*, *ermB*, *drfG*, and *vanHBX*, whereas *E. faecalis* contained *ant(6)-Ia*, *aph(3')-III*, *lsaA*, *ermB*, *cat(pC233)*, and *vanHBX*. These findings are consistent with the multidrug-resistant phenotype observed in the MIC assays.

- Correlation with MIC results: Importantly, the genomic findings correlate with the phenotypic susceptibility data. For example, the presence of vanHBX operons explains the high-level vancomycin resistance in both isolates, while ermB correlates with macrolide resistance, and aminoglycoside-modifying enzymes (aac(6′)-II, aph(3′)-III) with gentamicin resistance. The data have been presented and discussed in the Table 1, Fig. 2, Results and Discussion section. **Page 6, line 98.**
- Virulence factors and biofilm-associated genes: Screening against VFDB database revealed genes associated with adhesins, pili/aggregation, biofilm formation, immune evasion, proteases/invasion, and toxins. *E. faecium* harbored adhesin genes (acm, ecbA, scm), the aggregation factor (fss3), and immune evasion gene (sgrA), whereas *E. faecalis* carried a broader set of virulence determinants, including multiple adhesins (ace, efaA, EF0485), pili genes (ebpA–C, srtC, EF0149, fss2–3, prgB/asc10, fss1), biofilm-associated factors (sprE, fsrA–C, bopD, EF0818), immune evasion (cpsA–K), proteases (gelE, EF3023), and cytolysin toxin genes (cylR1–R2, cylA, cylS–M). This may explain the stronger biofilm formation observed in *E. faecalis* despite both strains being vancomycin-resistant. **Fig. 2; Page 6, line108; Page 12, line 243.**
- Prophage content: PHASTER analysis identified 1 intact prophage region in *E. faecium* and 2 intact prophage regions in *E. faecalis*. However, one caveat here is that we only used Illumina sequencing for these strains, still resulting in multiple contigs. Therefore, the number of intact phages might be an underestimation, since these still might be split over different contigs. Therefore, we did not include extensive description of prophage regions in the manuscript, since this is outside the scope of our study. **Fig. 2**

2. Phage Propagation and Purification

The manuscript lacks sufficient detail on the methods used for phage propagation and purification. This information is essential for reproducibility and should be clearly described.

Author’s reply: We thank the reviewer for this important remark. To improve reproducibility, we have expanded the description of our phage propagation and purification protocol in the Methods section. The revised text now provides a step-by-step description, including host preparation, infection conditions, clarification, and PEG/NaCl precipitation. **Page 18, line 370.**

3. Section: "Antibiotics Alone and Phage-Antibiotic Combinations Against Biofilm"

It is unclear whether phages were diluted in Mueller-Hinton Broth (MHB). The authors should also clarify what procedures were followed after the 4-hour incubation period.

Author’s reply: We thank the reviewer for this helpful request for clarification. In the revised Methods section, we have now explicitly stated the medium and the post-incubation procedures. **Page 20, line 420.**

4. Genomic Analysis Tools

The genomic analysis section does not provide a comprehensive list of the bioinformatic tools used. All software, databases, and parameters should be specified for transparency and reproducibility.

Author’s reply: We thank the reviewer for highlighting this important point. In the revised manuscript, we have expanded the Methods section to provide a detailed list of all bioinformatic

tools, databases, and parameters used in the genomic analyses to ensure transparency and reproducibility. **Page 17, line 360.**

5. Figure 1 - TEM Images

The quality of the TEM images is not suitable for publication. The presence of empty capsids and distorted (elongated) phage particles detracts from the figure's scientific value. Moreover, the figure legend is incomplete and should be expanded to fully describe the images.

Author's reply: We thank the reviewer for the constructive suggestions for improving Figure 1. We have revised the figure accordingly. Panels have been re-cropped so that the phage particles are centered and shown at larger size, highlighting their structural features more clearly. The scale bar has been increased in size for better readability. We adjusted the layout to provide a more balanced and visually accessible figures, ensuring that the phages are the central focus. Additionally, the figure legend has been expanded to provide a more complete description of the images, including the phage type, preparation method, staining procedure, magnification, and scale bar. **Fig. 3**

6. Phage Efficacy Testing

The authors should comment on the use of high phage titers in efficacy assays. The relevance of such high concentrations should be discussed in the context of potential therapeutic applications.

Author's reply: Thank you for this valuable comment. We agree that the use of high phage titers in efficacy assays requires careful discussion. In our study, we applied elevated phage titers to ensure reproducible biofilm penetration and to establish proof-of-concept for antibiofilm efficacy. It is well recognized that biofilm-embedded bacteria exhibit reduced phage accessibility compared to planktonic cells, necessitating higher initial titers to achieve measurable effects in vitro. From a translational perspective, the concentrations tested in vitro may not directly correspond to clinically achievable titers. However, the rationale for high input doses lies in mimicking potential local applications, such as intra-articular or topical phage administration, where high local titers can be reached without systemic toxicity. Several preclinical and clinical case studies have already demonstrated that local delivery routes (e.g., intraoperative irrigation, phage-loaded biomaterials, or catheter instillation) can achieve phage concentrations comparable to or even exceeding those tested in our assays. Thus, while our efficacy data reflect optimized in vitro conditions, they provide relevant insight into the therapeutic potential of phages, particularly when considering local, high-concentration applications in periprosthetic joint infections. Future studies are needed to define the minimal effective dose for clinical translation and to balance efficacy with practical manufacturing and delivery considerations. **Page 13, line 270.**

7. Lines 264-273 - Clinical Relevance of 100× MIC

The use of 100× MIC concentrations raises concerns about clinical applicability. While calorimetry data show significant differences, the authors should also attempt to estimate the actual bacterial reduction to contextualize the results biologically.

Author's reply: Thank you for this important comment. We acknowledge that the use of very high antibiotic concentrations (up to 100× MIC) raises questions regarding clinical translation. The rationale for testing these concentrations was to explore the upper limit of antibiotic activity in a biofilm setting, since biofilm-embedded bacteria are known to tolerate substantially higher concentrations than planktonic cells. We have added a clarification in the discussion to emphasize

that while 100× MIC conditions may not be achievable through systemic administration, they may be relevant in the context of local delivery strategies (e.g., antibiotic-loaded spacers, beads, or bone cement), where such high concentrations can be attained at the infection site without systemic toxicity. Regarding bacterial reduction, the calorimetry assay provides an indirect measure of bacterial metabolic activity rather than colony counts. Nevertheless, based on previously established correlations between calorimetry signals and bacterial load, a significant prolongation of time to detectable heat flow (tMax) at 100× MIC suggests at least a several-log reduction in metabolically active bacteria (doi: <https://doi.org/10.1016/j.resmic.2018.05.010>). This aligns with prior biofilm studies showing that antibiotics often require concentrations far above MIC to achieve even modest reductions in viable biofilm bacteria. **Page 10, line 206.**

Minor comments

1. Abstract

The phrase "particularly" may be misleading, as it implies specificity. Since all antibiotics showed activity, a more neutral phrasing is advisable.

Author's reply: We thank the reviewer for this helpful remark. The abstract has been accordingly revised.

2. Introduction

Line 66: Clarify why the strain is referred to as "unique." What distinguishes it from other strains?

Author's reply: We thank the reviewer for this comment. Accordingly, we replaced the vague word "unique" with a more precise description in the revised Introduction.

3. Lines 78-79

The statement does not clearly relate to VRE specifically; consider rephrasing or supporting with a citation.

Author's reply: We thank the reviewer for this valuable observation. The original statement has been revised to more clearly relate to VRE.

4. Materials and Methods

Line 100: The authors state that the phages are strictly lytic. Was this verified experimentally, such as through transduction assays?

Author's reply: We thank the reviewer for this comment. The whole genome sequencing showed a strictly lytic profile with no lysogeny-associated, virulence, or antibiotic resistance genes. The lytic nature of this phage was further validated using Phage.AI. **Page 7, line 128; Page 7, line 134.**

5. Line 109

The subsection on antibiotics should be formatted as a separate paragraph for clarity and better structure.

Author's reply: We thank the reviewer for this comment. The subsection on antibiotics has been formatted as a separate paragraph. **Page 18, line 391.**

6. Strain Selection

The rationale for choosing only two strains-one clinical isolate and one from a culture collection-needs further justification. How representative are these strains of the broader VRE population?

Author's reply: We appreciate the reviewer's concern regarding the representativeness of our strain selection. In fact, our study was initially designed with a broader panel VRE, including four *clinical E. faecium* isolates, one clinical *E. gallinarum* isolate, and the reference *E. faecalis* VRE strain ATCC 51299. However, during phage isolation from hospital sewage samples, lytic phages could only be recovered against one of the *E. faecium* clinical isolates and against *E. faecalis* ATCC 51299. Despite multiple enrichment attempts, no active phages were obtained for the other isolates. For this reason, our experimental work focused on these two strains, as they were the only ones for which we could identify strictly lytic phages. While we acknowledge that two strains cannot capture the full genetic and phenotypic diversity of the VRE population, our approach demonstrates the feasibility of isolating therapeutic phages and provides a proof-of-concept for their antibiofilm activity. Future work will expand strain coverage to include additional clinical isolates, aiming to identify phages with broader host range or to construct phage cocktails better representing the diversity of VRE. **Page 17, line 351.**

7. Lines 196-198

Which tool was used to suggest the creation of a new phage species? Was Phage AI or another platform employed? The basis for this conclusion should be explicitly detailed.

Author's reply: We thank the reviewer for this comment. VIRIDIC was used to calculate the intergenomic distance of our phages to the closest similar phages in the NCBI database. These results were included in Supplementary Fig. S1 and in the text, showing CUB-FS has <95% similarity to its closest relative. This criterion is defined as the threshold for a new species by ICTV. **Page 18, line 388.**

8. Figure 2: The current visual presentation makes it difficult to discern significant differences between groups. Consider improving contrast, layout, or including statistical annotations.

Author's reply: We thank the reviewer for this comment. We have restructured Figures 3 and 4 into Figure 3 including bar graphs which show the Δt_{Max} and ΔH_{FMax} relative to the growth control for each condition. This allows clearer visual comparison between treatments.

9. Figures 3 & 4: Legends should be more descriptive and provide enough detail to allow interpretation without referring back to the main text.

Author's reply: We thank the reviewer for this comment. All Figure legends in the manuscript have been revised providing enough detail to allow interpretation without referring back to the main text.

10. Discussion - Role of Phages vs. Antibiotics

The discussion mentions that certain antibiotics perform well against biofilms. If so, the rationale for introducing phages needs to be better articulated. What added value do phages provide in this context?

Author's reply: We thank the reviewer for this important point. The manuscript has been revised accordingly. **Page 12, line 235.**

Re: Spectrum01818-25R1 (Novel lytic phages improve antibiofilm activity of dalbavancin, daptomycin and fosfomycin against vancomycin-resistant enterococci)

Dear Dr. Svetlana Karbysheva:

Your manuscript has been accepted, and I am forwarding it to the ASM production staff for publication. Your paper will first be checked to make sure all elements meet the technical requirements. ASM staff will contact you if anything needs to be revised before copyediting and production can begin. Otherwise, you will be notified when your proofs are ready to be viewed.

Sincerely,
Olaya Rendueles
Editor
Microbiology Spectrum

Reviewer #1 (Comments for the Author):

The manuscript is much improved with much more appealing and clearer to understand figures. The authors have added clarifications in the text where necessary. There's a small mistake remaining: last figure says „legend" in legend. please fix.